

# Identifying agricultural pesticides that may pose a risk for birds

Cannelle Tassin de Montaigu and Dave Goulson

School of Life Sciences, University of Sussex, Falmer, East Sussex, UK

## ABSTRACT

In this study, we analyze changing patterns of pesticide use in agriculture in Great Britain over the 1990–2016 period, with respect to the risk they pose to birds. The weight of pesticides applied decreased by 51% between 1990 and 2016, but the area treated increased by 63% over the same period. Over this period, there has been considerable turnover in the pesticides used. The European Union (including Great Britain until 2020) has restricted or banned many pesticides for agricultural use, including organophosphates and carbamates. However, new generations of active substances have been introduced, such as the neonicotinoids, some of which have since been banned. In this analysis, we estimate the annual 'toxic load' of agricultural pesticide use in Great Britain for birds, measured as the total number of LD50 doses for corn buntings, *Emberiza calandra*. We have previously performed similar analyses for bees, for which the total toxic load increased six-fold during this period. In contrast, for birds the total toxic load fell by 80.5%, although still correspond to $8.3e^{+11}$ corn bunting LD50 doses in 2016. The decrease in toxicity is largely due to declining use of highly toxic organophosphates in recent years. We identify the pesticides in current use that may pose the highest risk to birds, which include a mix of insecticides, herbicides, fungicides, molluscicides, acaricides and plant growth regulators. The insecticide ethoprop was ranked highest in 2016, with a toxic load of 71 billion potential corn bunting kills. Some of the other chemicals presenting a high toxic load, such as the herbicide chlormequat, are not highly toxic to birds (in terms of LD50) but are used in very large quantities. However, it is important to stress that, in reality, only a tiny proportion of pesticides applied will be ingested by birds, and this will vary according to timing and method of application, persistence of the active substance and many other factors. We further note that impacts of pesticides on birds might often be indirect, for example via depleting their food supply, and that sublethal impacts may occur at much lower doses than the LD50, neither of which do we investigate here. Nonetheless, we suggest that this is a useful approach to highlight pesticides that might be worth closer study with regard to possible impacts.

## INTRODUCTION

It is often argued that agricultural intensification is necessary in order to overcome the challenges of food production for a rapidly growing human population

Corresponding author
Cannelle Tassin de Montaigu,
ct430@sussex.ac.uk

(*Godfray et al., 2010*). Artificial fertilizers and pesticides are an integral part of most intensive farming systems, but their use can be damaging to the environment. Since the publication of Rachel Carson's iconic book "Silent Spring" in 1962, the use of pesticide has been the subject of great controversy (*Carson, 1962*), and regulatory processes have been introduced to attempt to minimize the associated harm. The regulation of plant protection products in the European Union (EU) is regarded as being one of the most rigorous in the world as they review and re-evaluate the safety of each chemical regularly and Maximum Residue Limits authorized are amongst the lowest in the world. In recent years, the EU has attempted to promote reductions in use of pesticides via use of Integrated Pest Management practices, although the effectiveness of these measures is unclear (*Handford, Elliott & Campbell, 2015*). It is widely perceived that present-day pesticides are safer than older products (*Dudley et al., 2017*), but approximately 300 agricultural pesticides remain in use in the EU and there is a growing realization that regulatory tests do not always identify the environmental harm that can accrue when pesticides are applied in large quantities at a landscape scale (*Milner & Boyd, 2017*). Although improvements to tests are being developed, the majority of regulatory tests tend to focus on short-term studies of a handful of test species, each exposed to just a single active ingredient at a time, whereas in the real world a huge range of organisms are chronically exposed to complex mixtures (*Dudley et al., 2017*; *Milner & Boyd, 2017*).

One of the first pesticides to be linked to environmental harm was the organochlorine insecticide dichlorodiphenyltrichloroethane (DDT), commonly used between the 1940s and 1980s. It was shown to cause eggshell thinning in raptorial birds (*Rattner, 2009*). International bans and restrictions on usage of DDT and related pesticides have since been put in place. Many organophosphate insecticides have also been banned because of concern over risks to human health. Later, neonicotinoid insecticides were introduced (in the 1990s), and have become very widely used (*Goulson, 2013*; *Thomison, Jeschke & Butzen, 2012*; *Yamamoto, 1999*). However, clear evidence has emerged that they are contributing to decline in wild bees, butterflies, and aquatic insects (reviewed in *Pisa et al. (2017)*, *Wood & Goulson (2017)*), and the three most widely used neonicotinoids were banned from agricultural use in the EU in 2018.

While the mode of action of pesticides on target pest species is often well understood, more needs to be done to understand secondary modes of actions and their effects on other wildlife such as birds. There are about 574 different species of bird in the UK, of which 4% are considered farmland birds (*RSPB, 2020*). The populations of many farmland birds decreased critically during the last century across much of Europe (*Tucker & Heath, 1994*; *Pain & Pienkowski, 1997*; *Gregory & Gaston, 2000*). Farmlands represent a breeding and wintering habitat for a wide number of bird species (*Tucker & Heath, 1994*; *Tucker, 1997*). Given that arable and horticultural farmland are treated with an average of 17 pesticide applications per year (*Goulson, Thompson & Croombs, 2018*), there is clearly scope for birds and pesticides to come into contact. Direct exposure could occur via consumption of the crop seeds and/or contaminated water. Following a spray event, dermal exposure could occur; pesticides may be absorbed through the skin or consumed during preening behavior (*Vyas et al., 2007*). Over the past decade, research has found a

range of lethal and sub lethal effects due to neonicotinoids and other pesticides. These range from reduced chick survival and egg fertilization rate to cases of lethal poisoning in the red-legged (*Alectoris rufa*) and grey partridges (*Perdix perdix*) (imidacloprid, thiram and difenoconazole, *Lopez-Antia et al., 2013*; carbofuran, *Ljubojevic, Pelic & Kapetanov, 2016*). Abnormalities in testis and smaller embryo have been found in Japanese quail, *Coturnix japonica* (clothianidin, *Tokumoto et al., 2013*). Weight loss, clutch size reduction, reduced chick survival, and diminished mobility were found in northern bobwhite quail, *Colinus virginianus* (imidacloprid, *Berny et al., 1999*; various pesticides, *Poppenga & Tawde, 2012*; various pesticides, *Mineau & Palmer, 2013*). In songbirds, studies have found impaired navigation and reduced fat stores in white-crowned sparrows, *Zonotrichia leucophrys* (imidacloprid and chlorpyrifos, *Eng, Stutchbury & Morrissey, 2017, 2019*); brood size reductions in yellowhammer, Emberiza citronella (various pesticides, *Boatman et al., 2004*); and in house sparrows (*Passer domesticus*), sperm density was reduced by 50% (acetamiprid, *Humann-Guilleminot et al., 2019*).

In birds, the risk evaluation of pesticides exposure is very largely based on studies of four large bird species: The mallard duck (*Anas platyrhynchos*); gray partridge (*Perdix perdix*); Japanese quail (*Coturnix japonica*); and the northern bobwhite quail (*Colinus virginianus*). It seems unlikely that these species are truly representative of the range of sensitivities found across bird species (*Gibbons, Morrissey & Mineau, 2015*; *Luttik, Mineau & Roelofs, 2005*). We know that smaller birds are inclined to be more sensitive to acute poisoning by pesticides than larger ones (*Peters, 1983*; *Mineau, Collins & Baril, 1996*), and therefore harm to songbirds is likely to be underestimated. Corn buntings (*Emberiza calandra*) were identified by *Lopez-Antia et al. (2016)* as being a suitable passerine species for the risk estimation of pesticides. Corn buntings have been declining steeply in Great Britain and Europe since the 1970s and the intensification of agriculture is thought to have been a major driver (*Tucker & Heath, 1994*; *Hagemeijer & Blair, 1997*). Here we estimate the risks to corn bunting (as a proxy for all small birds) posed by changing patterns of pesticide use in farming in Great Britain, using the same method as *Goulson, Thompson & Croombs (2018)* and *DiBartolomeis et al. (2019)* who both focussed on honey bees (*Apis mellifera*). We quantify changes in the quantity of pesticides used, the area sprayed, and the total number of birds that could potentially be given a lethal dose each year over the 1990–2016 period, in the unlikely event that all pesticides were consumed by corn buntings. The Food and Environment Research Agency (*Department for Environment, Food and Rural Affairs (Defra), 2020*) provides detailed pesticide usage data across Great Britain both in terms of mass applied each year and area treated. This allows us to identify which pesticides used between 1990 and 2016 have the highest toxic load and thus may be deserving of closer investigation with regard to the risk posed to birds.

## MATERIALS AND METHODS

From all 424 pesticides listed on the DEFRA website, we removed those where data were unavailable and those for which the total usage over the 1990–2016 survey period was
below 100 kg, leaving 383 chemicals. Our analysis is based upon LD50 data for birds for each chemical from the existing literature. Since the LD50 only captures mortality, our approach does not assess sublethal effects or indirect effects such as food depletion. When LD50 data was available for multiple bird species, we choose in priority order, the value for the northern bobwhite quail, Japanese quail, red-legged partridge, and then mallard duck. If an LD50 was available for none of these species, we used whatever was available. When the available LD50s were greater than a certain value (e.g., >2,000 mg/kg of body weight), the > sign was simply removed. This slightly overestimates toxicity, but such pesticides are essentially non-toxic, so this assumption makes negligible difference to the overall calculations. For 89 pesticides, no LD50 values were available for any species of bird, so these were excluded. Ultimately, this left us with 294 chemicals in the analysis. The proportion of weight applied (in kg) for those 294 chemicals in 2016 constitute 94.3% of the total weight of pesticides applied in the UK (*Department for Environment, Food and Rural Affairs (Defra), 2020*). This suggests that our analysis only excludes pesticide used in small quantities.

We chose to conduct our analysis on the corn bunting as a reference passerine species, but we collected LD50 from a range of bird species. *Mineau, Collins & Baril (1996)* and *Mineau et al. (2001)* calculated scaling factors based on weight to refine extrapolation of pesticide toxicity in birds. In 1996, they calculated factors for 36 pesticides and found the average slope to be 1.15 and in 2001, they calculated factors for 130 pesticides with an average slope of 1.239. we used the average slope of 1.239 in our analysis, since this is based on a higher number of pesticides. To calculate the LD50 of the corn bunting for each chemical, we used the following equation:

$$\text{Log}(\text{LD50}_2) = \log(\text{LD50}_1) + (\log(W_2) - \log(W_1)) \times 1.239$$

where $\text{LD50}_1$ and $W_1$ are the LD50 and reference weight of the species available in the literature, and $\text{LD50}_2$ and $W_2$ the estimated LD50 and reference weight reference of the corn bunting. For example, for ethoprop, the LD50 for house sparrow is 0.1536 mg/bird (4.8 mg/kg of body weight in the literature), reference weight is 32 g for the house sparrow and weight reference for corn bunting is 49 g. This gives a LD50 estimate for corn bunting of 0.3 mg/bird.

We then calculated the number of corn bunting LD50 doses applied every year for each compound. This is calculated by the mass of this chemical applied divided by the LD50 (in mg/bird). This translates as the number of birds that could potentially be given an LD50 dose in mg/bird of each chemical, with the supposition that it was ingested by birds in its totality (in reality, a small percentage of pesticides used will interact with non-target species, perhaps none). We do not know the proportion of each compound that will be exposed to corn buntings or other non-target birds; therefore, it is important to restate that this is not an attempt to estimate actual bird deaths.

## Statistical analysis

All analyses were performed using R v. 3.1.1 (*Ihaka & Gentleman, 1996*). We ran linear models (LMs) with significance level set at 0.05 to test if pesticide toxic load were

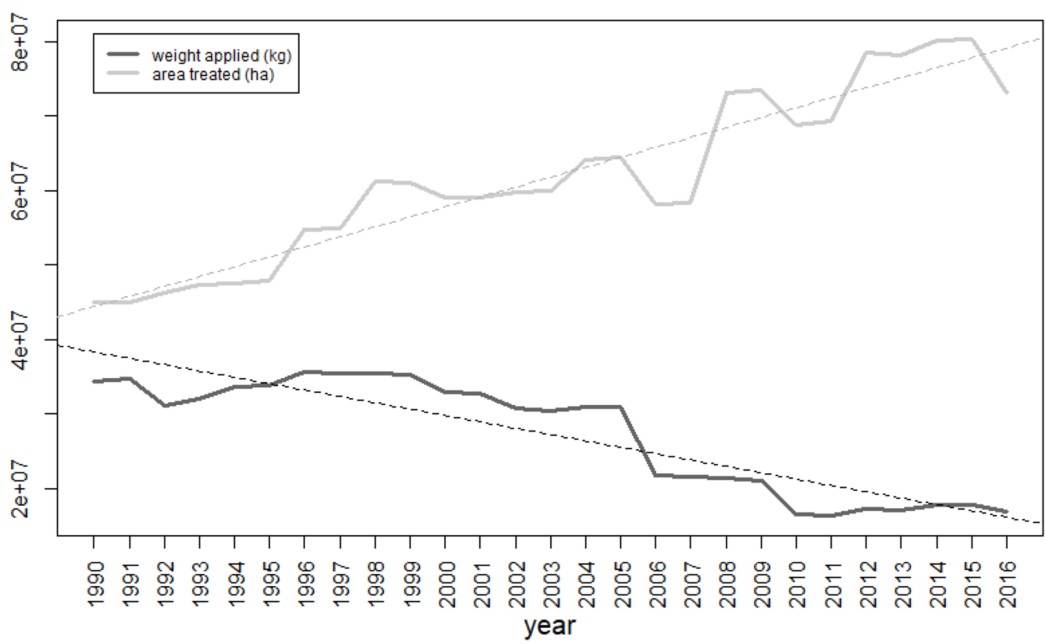

**Figure 1 Weight applied of pesticides (in kilograms, dark gray) and area treated (in hectares, light gray) in Great Britain between 1990 and 2016.** Fitted lines are the regression lines from the linear models.

significantly increasing or decreasing over the 1990–2016 period. Normality and homoscedasticity of residuals were verified by using quantile-quantile plots and the residuals plotted against predicted values.

## RESULTS

Between 1990 and 2016 the total weight of pesticides applied in Great Britain dropped by 50.9% (Fig.1; $F_{1,25} = 107.5$, $p < 0.001$). In contrast, the number of hectares treated increased by 62.6% between 1990 and 2016 ($F_{1,25} = 202.4$, $p < 0.001$). According to the Department for Environment, Food and Rural Affairs (*Department for Environment, Food and Rural Affairs (Defra), 2020*) the area of cropped lands stayed stable at around 4.6 million hectares during this time. Therefore, each hectare of crop received an average of 7.5 kg in total of plant protection compounds delivered in 1990 in 9.8 applications, vs 3.7 kg in 2016 delivered in an average of 15.9 applications. Overall, the number of corn bunting LD50 doses applied in the UK decreased over the last 27 years by 80.5% and represent on average more than $2.9e^{+12}$ corn bunting LD50 doses applied per year (Fig. 2). Toxic load due to insecticides, herbicides, fungicides and "other" types of pesticide all significantly declined between 1990 and 2016 (Fig. 3; insecticides: $F_{1,25} = 199.6$, $p < 0.001$; herbicides: $F_{1,25} = 125.4$, $p < 0.001$; fungicides: $F_{1,25} = 37.4$, $p < 0.001$; other pesticides: $F_{1,25} = 6.8$, $p < 0.05$). In 2016, insecticides represent 47% of the total toxic load applied in Great Britain, while herbicides, fungicides and "other" pesticides represent 22%, 13%, and 18% respectively. Toxic load due to insecticides decreased, largely due to the decreasing use of organophosphate insecticides, which are highly toxic to birds. They were largely replaced by pyrethroids, which are less toxic. Altogether, 9 insecticides, 52 herbicides,

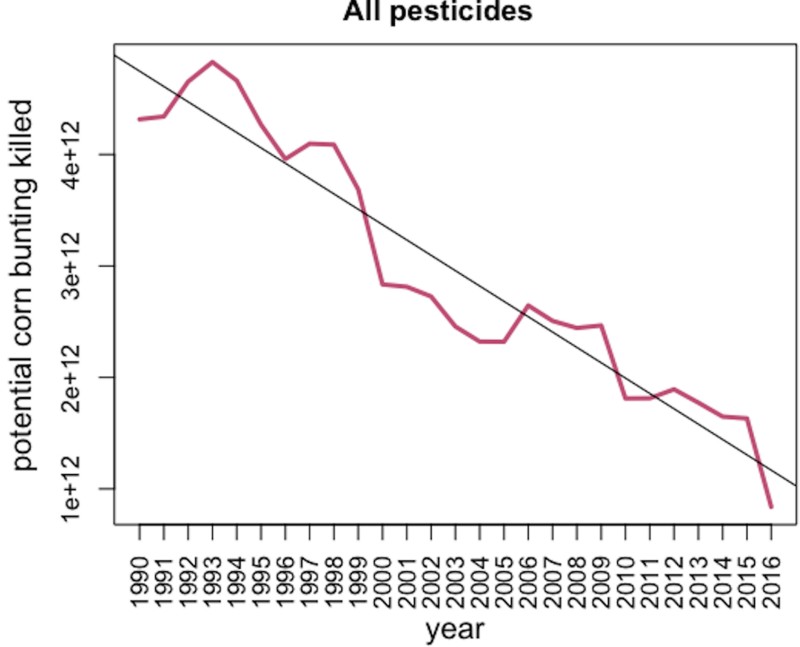

**Figure 2 *Potential* kill of corn bunting due to the application of the 294 pesticides included in the analysis in Great Britain between 1990 and 2016.**

36 fungicides and 12 "other" pesticides increased in terms of weight applied and hence their toxic load increased.

Although insecticides such as ethoprop have the highest toxicity to birds (very low LD50), the pesticides with the highest toxic loads included diverse chemicals, including plant growth regulator, herbicides, fungicides, and molluscicides (Table 1). This, in part, reflects the much higher application rates of some of these chemicals. Of the top 15, 7 significantly increased in use, and 2 significantly decreased over the period 1990–2016 (Table 1). Ethoprop had the highest ranked toxic load of any pesticide in 2016, representing $1.44 e^{+11}$ potential corn bunting LD50 doses in 2016 (Fig. 4), the result of quite large quantities being applied (43.2 tonnes in 2016) and a high acute toxicity to birds (LD50 for house sparrow of 4.8 mg/kg of body weight).

The plant growth factor chlormequat ranked second in terms of overall toxic load in our analysis (Fig. 5) and the herbicide pendimethalin ranked fifth with its toxic load to corn bunting significantly increased between 1990 and 2016 ($F_{1,25} = 124.8$, $p < 0.001$).

# DISCUSSION

Overall, we found that the toxic load due to insecticides, herbicides, fungicides and 'other' pesticides (e.g., molluscicides, plant growth factors, acaricides and others) were all decreasing through time, which is encouraging. In general, insecticides may be assumed to pose the highest risk to non-target animals due to their high toxicity, but our analysis suggests that this may not be true. We found that the top 15 pesticides with the highest toxic load in 2016 included 4 insecticides, 6 herbicides, 3 fungicides, 1 molluscicide and 1 plant growth regulator. This largely reflects the much higher application rates of

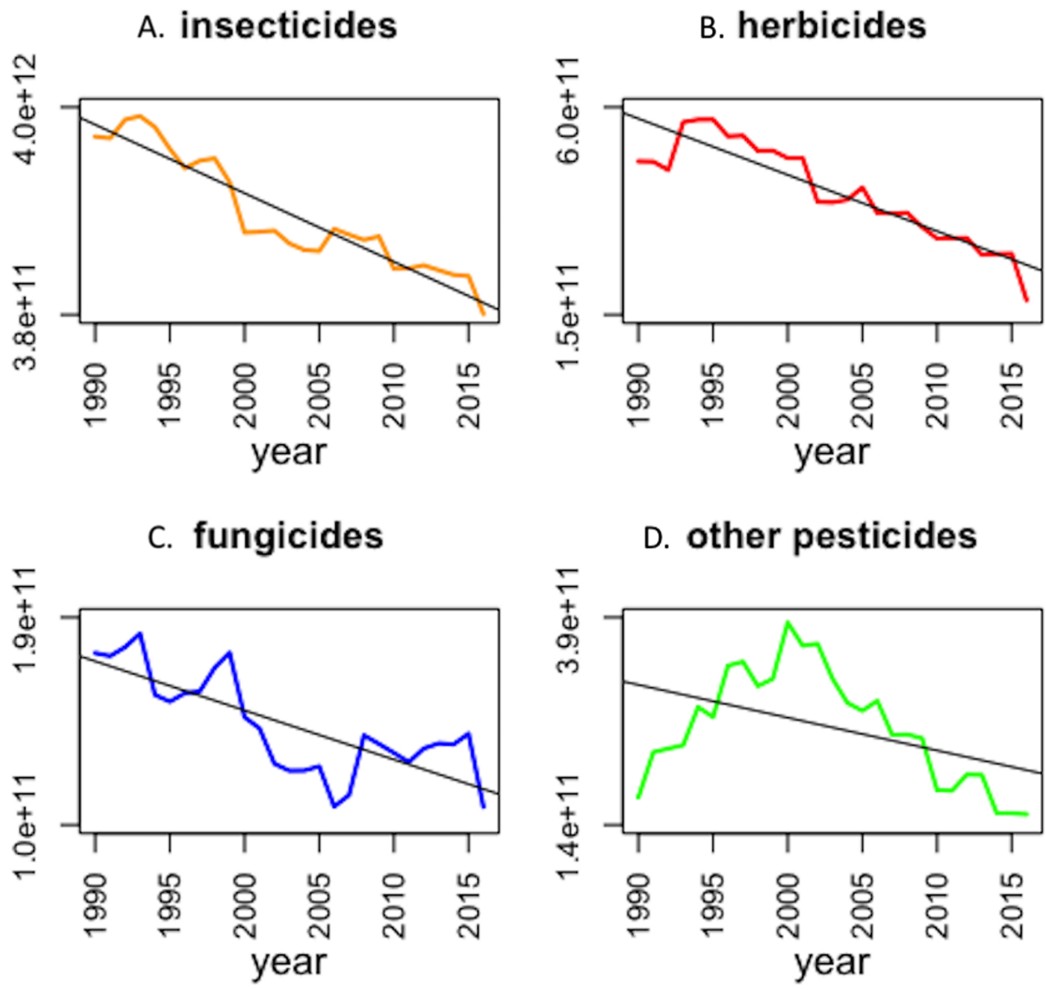

**Figure 3 Toxic load with respect to corn buntings, for all types of pesticides ((A) insecticides; (B) herbicides; (C) fungicides, and (D) other pesticides), applied in Great Britain between 1990 and 2016.** Fitted lines are the regression lines.

weed killers and fungicides, compared to insecticides. These numbers are, again, number of corn bunting LD50 doses applied and in reality, the large proportion of the compounds will be dispersed in the environment and just a very small proportion (perhaps none) will come into contact with birds. Many factors play a role in determining if and when a pesticide comes into contact with a bird, such as what crops it is applied to, at what time of the year, the method of application (e.g., granule, spray, pesticide-coated seed etc.) and its persistence in the environment. It may be possible to refine our crude analysis by attempting to incorporate such factors, for example by weighting pesticides according to their environmental persistence, or according to the abundance of birds in the crops to which they are typically applied and at the time of year when each chemical is usually used.

The pesticide with the highest toxic load from our analysis was ethoprop, an organophosphate insecticide-nematicide widely used in agriculture since it was first registered in 1966. Like all organophosphate pesticides, it inhibits the activity of

**Table 1 Summary of the 15 pesticides with the highest toxic load to corn buntings.** Arrows represent if the pesticide is increasing or decreasing significantly (in red) or just a tendency (in black).

| Rank | Type | Pesticide | Toxic load in 2016 | % Contribution to the total toxic load in 2016 | F value | $R^2$ | P-value | |
|---|---|---|---|---|---|---|---|---|
| 1 | Insecticide | Ethoprop | 1.44E+11 | 17.2 | 2.85 | 0.1 | 0.104 | ↗ |
| 2 | Plant growth factor | Chlormequat | 1.25E+11 | 14.9 | 0.19 | 0.008 | 0.663 | ↘ |
| 3 | Insecticide | Fosthiazate | 1.23E+11 | 14.6 | 81.61 | 0.77 | <0.001 | ↗ |
| 4 | Insecticide | Chlorpyrifos | 67,230,000,000 | 8 | 0.63 | 0.02 | 0.436 | ↘ |
| 5 | Herbicide | Pendimethalin | 41,391,128,049 | 4.9 | 124.8 | 0.83 | <0.001 | ↗ |
| 6 | Insecticide | Oxamyl | 39,405,000,000 | 4.7 | 14.66 | 0.37 | <0.001 | ↗ |
| 7 | Herbicide | Glyphosate | 30,774,835,165 | 3.6 | 137.9 | 0.85 | <0.001 | ↗ |
| 8 | Fungicide | Chlorothalonil | 24,272,184,179 | 2.8 | 31.05 | 0.55 | <0.001 | ↗ |
| 9 | Herbicide | MCPA | 20,512,500,000 | 2.4 | 63.14 | 0.72 | <0.001 | ↘ |
| 10 | Molluscicide | Metaldehyde | 18,845,000,000 | 2.2 | 1.358 | 0.052 | 0.255 | ↗ |
| 11 | Fungicide | Mancozeb | 17,454,837,310 | 2 | 42.1 | 0.63 | <0.001 | ↘ |
| 12 | Herbicide | Mecoprop-P | 15,811,868,132 | 1.8 | 0.361 | 0.014 | 0.553 | ↘ |
| 13 | Fungicide | Cyproconazole | 13,780,588,235 | 1.6 | 34.36 | 0.58 | <0.001 | ↗ |
| 14 | Herbicide | Prosulfocarb | 10,147,289,377 | 1.2 | 55.06 | 0.69 | <0.001 | ↗ |
| 15 | Herbicide | Diquat | 9,622,000,000 | 1.1 | 0.157 | 0.0062 | 0.695 | ↗ |

**Figure 4 Potential corn bunting killed due to the application of ethoprop (organophosphate insecticide) in the UK between 1990 and 2016 with regression line.**

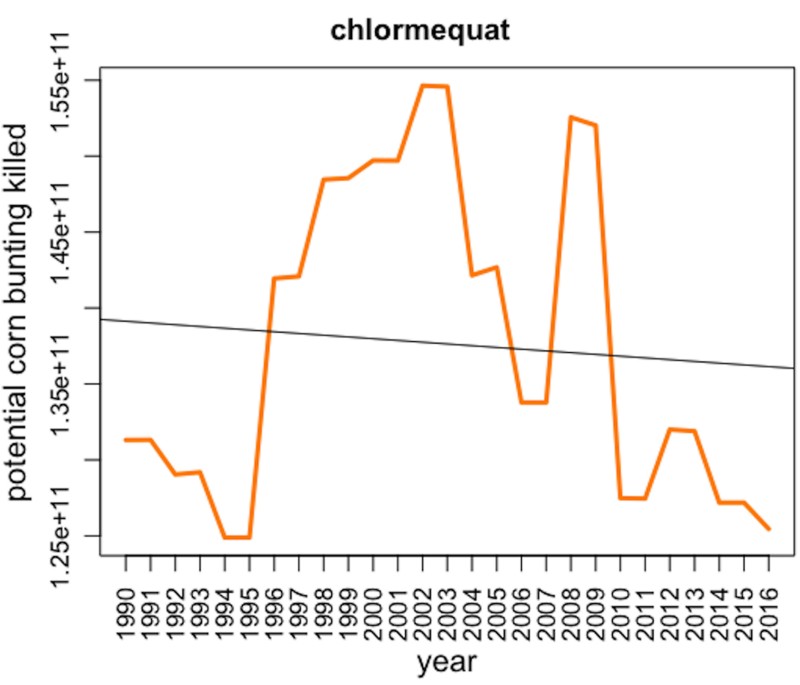

**Figure 5 Corn bunting *potentially* killed by the use of the plant growth regulator chlormequat over the period 1990–2016 in Great Britain with regression line from the linear model.**

cholinesterase. Ethoprop is usually applied either as a spray or in granular form. In the UK, the majority of its application is on potatoes (*Department for Environment, Food and Rural Affairs (Defra), 2020*). A total of 43.2 tonnes of ethoprop was applied to 7,736 ha in Great Britain in 2016 alone. Ethoprop is highly toxic to birds, with an LD50 of 4.8 mg/kg for the house sparrow. The European Food Safety Authority (EFSA) identified the ingestion of contaminated earthworms and other soil organisms, as well as the consumption of granules to be the main routes of exposure for birds (*European Food Safety Authority, 2006*). Ethoprop was identified as being responsible for the death of nine Canada geese *(Branta canadensis)* in Georgia (*Hill, 1975*; *Hudson, Haegele & Tucker, 1979*). Furthermore, a dunnock (*Prunella modularis*) carcass was found with ethoprop metabolites in its liver, although it is not clear whether this was the cause of death (*European Food Safety Authority, 2013*). Additionally, a variety of sublethal effects have been observed in birds such as: egg laying reduced by 22% in northern bobwhite quail treated with 7.5 ppm dose of ethoprop (*European Food Safety Authority, 2006*); hypothermia and tachycardia significantly different between pre-exposure and post-exposure (up to 6 h) in quails (*Heffernan, 2012*); decreased weight in hens fed 0.3 mg/kg/day over the 90 days of treatment (*Francis, Metcalf & Hansen, 1985*). It has also been demonstrated that ethoprop was found to be more toxic to birds when applied dermally than when given orally (*Hudson, Haegele & Tucker, 1979*; *Francis, Metcalf & Hansen, 1985*). Due to the accumulation of evidence on the lethal and sublethal impact of ethoprop on wildlife, in 2019 and with a maximum period of grace until March 2020,
the European Commission voted not to renew the approvals (*European Commission, 2020*), although it is still used in many other countries.

In second place in our analysis, chlormequat is a plant growth regulator widely used to reduce stem length in cereals such as winter and spring wheat and barley. The application of chlormequat has declined over time, but nonetheless 2,345 tonnes were applied in Great Britain in 2016. *Bro et al. (2015)* identified chlormequat as a likely contaminant of partridge egg clutches based on timing and method of application, but they do not directly quantify exposure. To the best of our knowledge, no field studies have looked at the impact of chlormequat on birds or other potential sublethal effects.

Ranked fifth in our analysis, pendimethalin is a dinitroaniline herbicide. More than 1,357 tonnes were applied in 2016 in Great Britain, against around 341 tonnes in 1990, representing a rise of about 297% in 27 years. As with chlormequat, no studies have attempted to quantify exposure of birds to pendimethalin, or to evaluate any impacts on health in the field.

Although we focus here on calculating potential direct mortality due to pesticides, indirect effects may contribute more to the decline of bird populations worldwide. Invertebrates represent a substantial portion of the food source of many vertebrate species including birds. During the breeding season, insectivorous but also primarily granivorous birds will raise their offspring with arthropods (*Cramp & Perrins, 1994*), and such effects cannot be captured by our analysis. Both insecticides and weed killers deplete the availability of prey insect species, the latter by reducing the abundance of non-crop plants which insects may feed on *Fuller (2000)*. Herbicides also eliminate weed species that herbivorous or granivorous birds may feed on *Boatman et al. (2004)*.

Studies have found that the availability of invertebrates used to feed corn bunting chicks on farmlands has decreased along with agricultural intensification and increase use of pesticides (*Brickle et al., 2000*). In the same study, they found that corn bunting nestlings were frequently fed grains, which it has been argued is a suboptimal food source used because insect food is unavailable. Limited food supply during the breeding season can lead to lower breeding success (*Brickle et al., 2000*; *Mullié & Brouwer, 1994*; *Teglhøj, 2017*). *Brickle et al. (2000)* estimate that this depletion of insect food could have contributed to the decrease in corn bunting and be retarding population recovery in the South Downs, Sussex (UK).

## CONCLUSION

We recognize that our crude analysis does not capture many of the complexities of the possible impacts of pesticides on birds. It suggests that the potential exposure in terms of toxic load of pesticides to birds has decreased in Great Britain over the past 27 years, mainly due to moratoria and bans by EU authorities, particularly the withdrawal of most organophosphate insecticides. Our analysis suggests that some groups of chemicals that generally receive little attention with regard to risks to birds should be given more scrutiny, including the plant growth regulator chlormequat, herbicides such as pendimethalin, fungicides such as mancozeb, and molluscicides such as metaldehyde.

## ACKNOWLEDGEMENTS

We are very thankful to the pesticide usage statistics team at the farmed environment research agency (FERA, *Department for Environment, Food and Rural Affairs (Defra), 2020*) who conducted the pesticide surveys, as well as the different articles and reports where LD50 values on birds were available.

### Funding

This work was funded by SongBird Survival. The funders had no role in study design, data collection and analysis, decision to publish, or preparation of the manuscript.

### Grant Disclosures

The following grant information was disclosed by the authors:
SongBird Survival.

### Competing Interests

The authors declare that they have no competing interests.

### Author Contributions

- Cannelle Tassin de Montaigu performed the experiments, analyzed the data, prepared figures and/or tables, authored or reviewed drafts of the paper, and approved the final draft.
- Dave Goulson conceived and designed the experiments, authored or reviewed drafts of the paper, and approved the final draft.

### Data Availability

  The raw data are available in the Supplemental Files.

### Supplemental Information

Supplemental information for this article can be found online at http://dx.doi.org/10.7717/peerj.9526#supplemental-information.

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
