# Peer review of "Identifying agricultural pesticides that may pose a risk for birds"

_PeerJ, doi:10.7717/peerj.9526_

## Round 0.1 · original submission · Major Revisions

As you will see from the results, several reviewers recommended minor revisions. However, a review from an expert in the field was more critical and outlined a number of shortcomings that must be addressed prior to consideration for publication.

Reviewer 1 ·

Basic reporting

Good reading and rational.

Experimental design

Descriptions are sound and good to follow.

Validity of the findings

The topic is appealing the authors present an interesting case. The general trends over such good length of time are invaluable. My main concern regards the generality of compounds used since pesticides are a rather diverse set of chemicals with different properties. However, the authors did track down the main compounds and their likely impact in the system considered recognizing the limitations of the study. My concern refers on the estimate of toxic load likely grossly over-estimating the studied effect. On the other hand the reliance on only LD50 and LD50 extrapolated from other species is also a key shortcoming of the method. Below I detail a bit more my concerns.

Additional comments

I liked the study and argument. The text is persuasive and engaging and the concept of “toxic load” explored is interesting. However, some of the values estimated sound strange and unrealistic – toxic load in the trillion range, but resulting in so few lethal responses (sublethal not even considered although they tend to be larger) is very hard to defend and even contextualize since is not particularly realistic. This is my key concern. The use of a general standard such a “toxic load” is useful and its relative determination helps, but the exaggeration distorts reality leading to bias. I would cautious against that. That said, the authors do make a clear point even at the Abstract, recognizing some important limitations of the approach, which is welcome.

Some minor points:

Lns 50-52: this is not a good reason to justify how rigorous is the EU legislation as virtually every country in the world requires regular re-assessments or reviews of registered chemicals for agriculture use. Thus, please revise providing a more compelling reason;

Lns 55-59: it is more than that, as even assessing impact lower amounts in complex system is also difficult. High or low are relative terms and depends on the context and ecosystem. Improvements are possible, perfection is not, but some longer ter-tests do exist and sublethal assessment and multiple-species assessments are also been considered for inclusion in protocols for registration;

Lns 59-62: The multiple chemical exposure is an excellent point!

Lns 75-77: The mode of action does not change with the species, but the secondary modes of action and their relevance may. Please consider reviewing for higher precision and clarify;

ln 84: prey consumption is not direct exposure;

ln 105-111: Although recognizing the relevance of the study, the sell of of a particular species as indicator or sentinel species based on alleged representativeness, I am skeptical on the subject and more in favor of broader surveys covering different species over time and recognized the most affected, if significant effect is detected in such assemblage-level studies.

Lns 115-116: the catch is really recognizing the rational behind the “potential killing” that serves as basis for the “toxic load” to prevent value overestimation;

Ln 129-130: LD50 is a sound toxicological endpoint, but lethality-based and not accounting for indirect effects nor sublethal effects, both of which are important and likely more important than lethality itself. This shortcoming need to be recognized upfront since it is a conceptual oversimplification;

Ln 141-150: methodological OK I guess, but subject to gross mis-estimations since the data is not really known of the species under investigation;

Ln 155-158: This is bound to drastically overestimate the effect pursued. This is so because a rather small percent of the applied compound will actually reach the target assessed up to the point of causing its death. The “potential” here is with great capital letters…I do like a standardize way to draw attention to the load applied and potential effect, but the distortion here bother me as a gross over-simplification piled over the the species extrapolation; I do appreciate that the intent is not estimate bird death, but still the grossly biased estimation is a concern since it is far from realistic and prone to generate the wrong kind of controversy.

Just an example of the shortcoming: if LD50 is considered, mortality estimates based on pesticide load applied should reflect the “kill” of 50% of the population, not 100%...And this is just one issue on such estimates;[of course it does not make much different when one talks about the trillion range…; still, trillion…]

The qualitative trends observed seem OK and intuitive matching my own experience. I also appreciate the fact that the authors were candid in recognizing some of the limitations of the crude analysis provided. However, I believe my concerns remains valid.

Reviewer 2 ·

Basic reporting

Generally well written article - a few small grammatical errors.

This reviewer would have liked perhaps more discussion around pendimethalin and possible adverse effects - while it might be limited, it would help to make the link between the results of the assessment of toxic load and real adverse effects.

Experimental design

This is an interesting methodology for assessing the impacts of pesticides on non-target species over time. There are obvious limitations to making specific conclusions on acute toxicity to non-target species - but the authors are careful to note the limitations. The methodology is solid given the limitations. Although the methodology related to extrapolation of LD50 values from other bird species to the bunting is clear, it might have been helpful to have provided an example calculation, say for chlorpyrifos or pendimethalin.

Validity of the findings

Within the limitations of the study, the approach provides an interesting way to assess the historical impacts of pesticide use on non-target species.

Additional comments

In the same vein as earlier comment about relevance of toxic load vs. real effects, it would be useful perhaps to tie results to specific actions that could be take to help better define true risk - e.g., what specifically should be done to further evaluate the true risk of pendimethalin use on non-target species.

Annotated reviews are not available for download in order to protect the identity of reviewers who chose to remain anonymous.

·

Basic reporting

Several factual errors and serious gaps in the review of the seminal literature lead me to believe that the authors do not have a good grasp of the subject - pesticide effects on birds. I have included some of those details in 4. below.

Experimental design

Given the title, I was looking forward to reviewing this paper. Unfortunately, I have serious reservations. Beyond some possible errors in the choosing of toxicity values for the different pesticides (see detailed comments below), I have a more fundamental problem with the approach and believe the conclusions are simply wrong-headed and misleading. Fundamentally, I do not agree with the authors’ conclusion that: “… this is a useful approach to highlight pesticides that might be worth closer study with regard to possible impacts.” By impacts here, the authors mean lethal impacts on birds - as predicted by a lethal-toxicity-to-bird measure. Avian LD50 and amounts of pesticides used are the only two variables in play.

Indeed, this conclusion is already negated by this cautionary statement – with which I do agree: “We do not know the proportion of each compound that will be exposed to corn buntings or other non-target birds, therefore it is important to restate that this is not an attempt to estimate actual bird deaths. (At line 158)”

I should point out that, despite this caution, “potential corn buntings killed” is the tabulated measure and the paper purports to indicate which of the registered pesticides present the highest risk to birds on that basis. But this is more than semantics, and simply finding a different name for the calculated ‘toxic load’ would not satisfy me. The problem is more fundamental.

This analysis styles itself after similar ones on the toxic potential of pesticides to bees (Goulson et al. 2018; DiBartolomeis et al., 2019). In defense of the ‘toxic load’ approach in bees, it could be argued that, bees being insects, most (all?) insecticides applied in fields at levels necessary to kill insect pests are indeed likely to be causing some degree of toxicity and mortality in bees. Therefore, translating weights of chemicals applied into potential lethal doses is not quite as problematic – not ideal but tenable. The same cannot be said for the application of the concept in birds. By simply totaling up the number of LD50s, a massively used pesticide of low to middling toxicity could return a ‘risk value’ that is as high or higher than an acutely toxic pesticide used only in small amounts; e.g. carbofuran which probably was excluded from the analysis because only a small amount was used in the UK historically. As used in this paper, the ‘toxic load’ approach is especially troublesome because all pesticides are included whether or not a toxic effect is even remotely possible. This includes herbicides (28.5% of the overall ‘toxic load’) and fungicides (15% of the toxic load). Some of these (e.g. pendimethalin – apparently the second-most dangerous pesticide to birds after chlorpyrifos), are used in large quantities but have never been shown to present any semblance of a direct risk to birds – even though lethal effects and debilitation are the only effects that can be predicted by an LD50 value.

I therefore do not agree that the method exemplified in this paper is a reasonable way to set priorities for looking at safety of pesticides to birds. Worse, the proposed method is likely to obscure which compounds do represent a real direct risk to birds – e.g some neonicotinoid seed treatments for which there already is field evidence.

I might be willing to believe the authors truly realise the limitations of their approach except that they do not seem to be aware of more serious attempts to quantify lethal bird effects as a function of pesticide toxicity. None of the key papers are cited. See for example my own contribution to the field:

“Mineau, P. 2002. Estimating the probability of bird mortality from pesticide sprays on the basis of the field study record. Environmental Toxicology and Chemistry 24(7):1497-1506.”
“Mineau, P. and M. Whiteside. 2006. The lethal risk to birds from insecticide use in the U.S. – A spatial and temporal analysis. Environmental Toxicology and Chemistry 25(5):1214-1222.”

I should note that the approach documented above was used in the EU to ‘calibrate’ lethal risk assessment calculations based on LD50; viz.:“Anonymous. 2008. Scientific Opinion of the Panel on Plant protection products and their residues on a request from the EFSA PRAPeR Unit on risk assessment for birds and mammals. The EFSA Journal (2008) 734, 1-181.”

By reviewing the various toxic effects of chlorpyrifos, the authors also seem to be suggesting that their measured toxic load says something about sublethal toxic effects of pesticides. Unfortunately, there is no good evidence for this. Many of the more 'interesting' effects of pesticides on embryonic viability, for example, do not correlate with toxicity. There are too many numerous examples to cite - starting with DDT which has a very low acute lethal toxicity to birds and other vertebrates!

The paper ends (Lines 254 onward) with a discussion of indirect effects of pesticides on birds – through insect removal. Although I do not disagree with the importance of insect removal for birds, this is – as stated by the authors – beyond the scope of this paper which looks at lethal toxicity to birds only as an endpoint. What I do object to is that the issue of indirect effects appears to be used here to justify the ‘toxic load approach’ espoused in the paper; e.g. why should pendimethalin have the second highest risk to birds after chlorpyrifos…. Viz: " ... studies on soil organisms found that pendimethalin reduced reproduction capability and number of offspring in springtails (Folsomia
250 candida), and reduced weight gain in earthworm (Eisenia fetida; Belden et al., 2005)" (At Line 248).

This is totally mixing apples and oranges and is nonsensical. The potential impact of pesticides to reduce the insect food supply of birds will look a lot more like the honeybee toxic load analysis – with a yet to be designed calculation of insect loss through vegetation control - a doubly-indirect effect. I am willing to bet that the results will look very different.

The issue of indirect effects is certainly no justification for the approach presented in this paper.

Validity of the findings

As outlined in 2. above, I do not believe that the findings have any validity and may seriously mislead and/or hamper the search for truly damaging pesticides; Viz: “Our analysis suggests that some groups of pesticides that generally receive little attention with regard to risks to birds should be given more scrutiny, including herbicides such as pendimethalin, fungicides such as mancozeb, and molluscicides such as metaldehyde." (At line 275)

Additional comments

Line 65. “(DDT) and its metabolite dichlorodiphenyldichloroethylene (DDE), a seed treatment commonly used between the 1940s and 1980s…”
I believe this is factually incorrect. DDT was not used for seed treatments – it certainly was not its main use. A number of cyclodiene organochlorine insecticides – e.g. Aldrin, dieldrin, chlordane – were used as seed treatment insecticides.
Lines 66-67. “It was shown to cause eggshell thinning in raptorial birds (Rattner, 2009) and several marine species were also lethally affected (Pašková, Hilscherová, & Bláha, 2011).”
I am assuming you mean marine bird species here? Otherwise, it is off-topic. If so, the statement is also largely incorrect. Eggshell thinning also extended to a number of marine species in order Pelicaniformes. Lethal effects with DDT were rare – only seen in cases of extreme starvation or in a few cases of very high application rates and earthworm contamination – e.g. American robins. Most of the lethal effects from organochlorines were from cyclodienes. It was the combined effect of DDE eggshell thinning and the acute toxicity of cyclodienes that proved catastrophic for a number of bird species.
By the way, I would argue that the effects of OPs on birds probably predated the DDE/eggshell story which took a while to unravel. Lethal effects on birds from OPs were documented very early indeed – see Silent Spring.
Line 74: ‘were’ banned
Lines 75-76. “While the mode of action of pesticides on target pest species is often well understood, little is known about their effects on other wildlife such as birds.”
Having published on the impacts of pesticides on birds for the last 40+ years, I would argue this is a ridiculous statement to be making as worded.
Lines 87-90. Not a proper sentence
Line 107. There is direct evidence in birds that size has an effect … but not how Peters envisioned it. Actually the opposite … so am not sure what part of Peters you are citing. See: “Mineau, P., B.T. Collins and A. Baril. 1996. On the use of scaling factors to improve interspecies extrapolation of acute toxicity in birds. Regulatory Toxicology and Pharmacology 24:24-29.” This is before any consideration of the importance of body size on food intake.
Aaah. I see you do cite that paper later on. So you are aware that scaling does not follow the mammalian 2/3 or ¾ rule as envisioned by Peters and which forms the basis of most mammalian toxicology extrapolations.
Line 130. “Whenever available, we used 48 h oral acute toxicity LD50 values”
There is no such test for birds. 48 h is often used in the aquatic field to denote exposure time for measurement of LC50s.
Line 134. “For 54 pesticides, we used an average of LD50s from multiple unspecified bird species given in Mineau et al. (2001).”
Actually, I do not provide average values in that publication. Are you using the median or the HD5? Also, what are you using for the other 240 chemicals in the analysis? – presumably a single value given that you apply bodyweight scaling!? If you read the paper you are citing (Mineau 2001), you will see why it is difficult to compare toxicity among compounds with differing amounts of test data. As described, the methodology used is not at all clear.
Line 144: “In 1996, they calculated factors for 36 pesticides and found the average slope to be 0.5mg/kg.”
The 0.5 mg/kg value is not a slope term …not sure where it comes from. In fact, in our 1996 sample, we obtained an average slope of 0.15. In the larger 2001 sample, we got 0.239. … Viz: “For the 130 pesticides with n ≥ 4, the regressions were positive in 99 cases and the overall mean slope was 0.239; this was slightly more extreme than the value of 1.15 (corresponding to 0.15 when expressed on the basis of mg/kg) reported by Mineau et al. for a subset of 36 pesticides.(Mineau et al. 2001)”

So … the methods section leaves me wondering what procedure was used when several LD50 values were available. How could you do a body weight correction for the 54 pesticides for which a (median?) value was obtained from our 2001 paper? Is this comparable to the values you generated for the remaining 240 pesticides? How so?
Line 224. “However, we cannot be sure that applications to farmland in the UK are causing harm to birds.”
There are a number of field studies that do show impacts from chlorpyrifos when used as a spray. If memory serves, the supplementary material reviewed in the 2002 article cited above should have those references. Those studies were also reviewed by EFSA in 2008.

---

## Round 0.2 · Minor Revisions

Originally I received three reviews, two of which recommended changes prior to acceptance, and one, Prof. Pierre Mineau, who recommended rejection. I believe that you recommended Dr. Mineau as a reviewer, and he provided the harshest review. Since two of the three reviewers felt that the manuscript was suitable with changes, I gave Dr. Mineau a chance to read the revision, but unfortunately he was not convinced and still feels that the manuscript should be rejected. He asked that you remove his name in your opening comments and acknowledgements as he felt that it would insinuate that he supported the paper, which he does not. Therefore, please revise accordingly. Upon resubmission, I will accept the paper given that two of the reviewers felt that the manuscript is suitable for publication, and I agree.

·

Basic reporting

.

Experimental design

.

Validity of the findings

.

Additional comments

My fundamental issues with the methodology persist. I tried to explain my thinking re. the difference between a 'toxic load' approach for insects exposed to insecticides only (our PLOS One paper) vs. birds being exposed to all sorts of products ... some of low toxicity but very high volume/popularity.

Unfortunately, this led to the following comment in your rebuttal statement: "We are somewhat puzzled by the reviewer’s negative response to this approach, since it is almost identical in approach to a paper Prof Mineau published in PlosONE last year" .... a comment I found a little disingenuous as I think a clear reading of my comments should have removed any puzzlement.

But I see I will not succeed in convincing you. I do believe you need to clarify your methodology however. As I pointed out, the largest degree of uncertainty comes from the wide variation in toxicity between species. The body weight scaling issue is a minor one in comparison. You deal with the latter - which is good - but it still is not clear what departure point you are using to set a toxicity value. In your rebuttal comments, you mention taking the bobwhite as the chosen species. This is not what the text says. In any case, using a single species approach brings much more uncertainty to the calculations. That was the reason behind our compendium and we provide the factors necessary for a (quasi) distributional approach where too few species have been tested... but it doesn't look like I will convince you of that either.

In any case, I would ask that you please remove the mention that I was a reviewer. This does give the impression of an endorsement.

---

## Round 0.3 · accepted · Accept

Thank you for the additional changes.